# Global sex differences in hygiene norms and their relation to sex equality

**Kimmo Eriksson**[1,2]*, **Thomas E. Dickins**[3], **Pontus Strimling**[1]

**1** Institute for Futures Studies, Stockholm, Sweden, **2** Division of Mathematics and Physics, School of Education, Culture and Communication, Mälardalen University, Västerås, Sweden, **3** Department of Psychology, School of Science & Technology, Middlesex University, Hendon, London, United Kingdom

* kimmoe@gmail.com

**Data Availability Statement:** The data analyzed here and the code for the analyses behind all tables and figures are all available at the Open Science Framework (osf.io/d4q38/).

## Abstract

Strict norms about hygiene may sometimes have health benefits but may also be a burden. Based on research in the United States, it has been suggested that women traditionally shoulder responsibility for hygiene standards and therefore tend to have stricter views on hygiene. However, there is little systematic research on sex differences in hygiene norms at the global scale. We set up two hypotheses: (1) Stricter hygiene norms among women than among men is a global phenomenon. (2) The size of this sex difference varies across nations with the level of sex equality. We examine these hypotheses using data from a recent international survey (N = 17,632). Participants in 56 countries were asked for their views of where it is not appropriate for people to spit and in which situations people should wash their hands. As a measure of sex equality, we use an existing country-level measure of attitudes to equality between the sexes, available for 49 nations in the study. Stricter hygiene norms among women than among men are observed almost everywhere, but there are a few exceptions (most notably Nigeria and Saudi Arabia). The size of the sex difference in hygiene norms varies strongly with the level of sex equality, but in a non-linear way. The sex difference is most pronounced in moderately egalitarian countries with the highest recorded difference being in Chile. In more egalitarian parts of the world, more sex equality is associated with a smaller sex difference in hygiene norms. In the less egalitarian parts of the world, the opposite relation holds. We offer an interpretation in terms of what different levels of sex equality mean for the content of sex roles.

## Introduction

Societies have norms about hygiene. In Europe, norms restricting spitting in public have been found in etiquette books dating back to the 16th century [1]. Since the adoption of germ theory in the late 19th century, hygiene behaviors such as hand washing and refraining from spitting have been commonly regarded as important means to control the spread of infectious diseases [2]. With the adoption of germ theory, successful anti-spitting campaigns were organized in the United States and Europe [3]. Norms about hygiene received renewed attention with the outbreak of the COVID-19 pandemic, with public health experts emphasizing the importance of increased hand sanitization (e.g., Dalton et al. [4]) and of reducing respiratory droplets by

**Funding:** KE acknowledges financial support from the Swedish Foundation for Humanities and Social Sciences (Riksbankens Jubileumsfond) (grant no. P17-0030:1). PS acknowledges financial support from the Knut and Alice Wallenberg Foundation (grant no. 2017.0257). The funders had no role in study design, data collection and analysis, decision to publish, or preparation of the manuscript.

**Competing interests:** The authors have declared that no competing interests exist.

using facemasks, sneezing, and coughing into the elbow, and refraining from spitting [5, 6]. Importantly, hygiene norms are not fixed. Elias [7] documented how hygiene norms in Europe have become increasingly stricter over centuries and proposed that this shift was due to an increasing societal valuation of self-control. Hygiene norms also vary across societies. In line with Elias' theory, a study of 56 countries found that societies that value self-control more highly tend to have stricter norms about hand washing and spitting [8]. Here we revisit this data to examine sex differences in hygiene norms.

There are indications that men tend to regard these hygiene behaviors as somewhat less important than women do. It has been anecdotally observed that spitting in the United States seems more common among men than among women [9], but hard evidence on spitting norms is absent. For hand hygiene, however, there is data available. A recent survey among American men and women [10] found that slightly fewer men than women rated hand washing as very important in several key contexts (84% vs. 91% after using the toilet; 68% vs. 72% before a meal; 66% vs. 74% after using public transport). This sex difference in hygiene norms is consistent with behavioral studies of hand washing not only in the United States but also in Egypt, Ghana, Hong Kong, and China [11–15]. However, there is little theoretical work on sex differences in hygiene norm preference. Note that this is not a difference in how norms say women and men ought to behave but a difference in how women compared to men say that everyone ought to behave.

The literature on sex differences has been divided between arguments for biological and social causes [16]. Within this literature disgust sensitivity has been inspected, with women repeatedly demonstrating greater disgust sensitivity than men [17–20]. Disgust responses are not unrelated to hygiene behaviors and are thought to protect individuals from risky exposures that will be detrimental to average lifetime inclusive fitness, although country measures of disgust sensitivity are not strongly related to pathogen prevalence [20]. The sex difference in sensitivity has been attributed to asymmetries in fitness costs across the sexes, for example linking the female response to factors such as the unique challenges of pregnancy and post-partum obligate care [17]. Sex differences in disgust cannot fully explain the emergence of sex differences in generic normative standards for two reasons. First, hygiene practices and hence hygiene beliefs are not all directly tied to perceptible stimuli that elicit disgust. This is particularly the case for norms around handwashing, which is a core measure in the current study. Second, while the propensity for having normative beliefs about what ought to be done is not historically recent and we might anticipate biological limits on the *types* of beliefs that can be held, changes in their content, in the *tokens* of belief, can only have been caused by external contingencies. Hygiene practices have developed in recent historical time and represent a change in external contingency. If disgust response, or any other biologically caused disposition plays a role, those aspects of the behavioral phenotype have nonetheless interacted with new and emergent aspects of the relevant social ecology [16]. A starting point for investigation would be to explore cultural variation in sex differences in normative beliefs.

One suggestion for why women would have stricter hygiene norm preferences comes from the work of Nancy Tomes. According to Tomes' [2] account of the development of new hygiene norms in late 19th and early 20th century America, the responsibility for hygiene standards tended to be placed on women. On this basis, it has been hypothesized that higher female strictness in hygiene reflects this asymmetry in responsibilities [21]. From this socio-historical perspective, stricter hygiene norms among women may be interpreted as a burden rather than an asset. The theory that sex differences in the strictness of hygiene norms are an expression of underlying inequality between the sexes yields specific predictions. Since inequality between the sexes that favors men is a global phenomenon, a straightforward prediction is that we should observe stricter hygiene norms among women across the globe.

**Hypothesis 1.** In all societies, women have stricter norms about hand washing and spitting than men do.

Since the level of inequality between the sexes varies dramatically across the globe, we should also observe variation between countries in the size of the sex difference in strictness of hygiene norms.

**Hypothesis 2.** The sex difference (i.e., how much stricter women's norms about hand washing and spitting are than men's) decreases with the level of sex equality in the society.

We are not aware of any prior study that addresses sex differences in hygiene norms in a global perspective. In the present study we examine our hypotheses by taking sex into account in a reanalysis of the aforementioned data on hygiene norms [8, 22]. We control for a couple of factors known to influence strictness of hygiene norms: valuations of self-control [7] and perceptions of a disease threat [23]. With respect to Hypothesis 1, we find stricter hygiene norms among women than among men *almost* everywhere, but in two countries (Nigeria and Saudi Arabia) we find significant sex difference in the reverse direction, that is, with looser hygiene norms among women than among men. With respect to Hypothesis 2, we find that sex difference in strictness of hygiene norms varies strongly with the level of sex equality, but in a non-linear way. When we look at the span from moderately egalitarian countries to the most egalitarian countries, the sex difference in hygiene strictness indeed decreases as predicted. However, in the span from the most inegalitarian countries to moderately egalitarian countries we find the opposite relation. In other words, the sex difference in hygiene strictness peaks in moderately egalitarian societies.

## Method

We focus on hygiene strictness in the sense of normative beliefs: how people think one *should* behave [8]. An advantage of belief measures compared to behavioral measures of hygiene is that they are less sensitive to the respondent's personal situation. For instance, some people may not have access to a public swimming pool or any interest in going to one, but they may still have an opinion on whether it is okay to spit in it.

The level of sex equality in a country can be operationalized in various ways. We are concerned with sex inequality in terms of ingrained differences in the treatment of men and women solely due to their sex. The best available measure of this, we believe, is measures of cultural values with respect to treating men and women equally or differently. As detailed below, the World Values Survey provides such a measure [24]. This is our key independent variable.

An alternative operationalization of sex equality is the Global Gender Gap Index (GGGI), which estimates national gaps in outcomes between the group of men and the group of women with respect to economic participation and wages, educational attainment, political empowerment, and health and survival. It has been argued that different levels of structural sex equality may result in the same outcome and therefore that the GGGI is unsuitable to represent sex equality as a causal factor [25]. Indeed, when recent research on sex differences in student achievement used both cultural values with respect to sex equality and the GGGI as independent predictors, only cultural values had an independent effect [26]. In the current study, we therefore operationalize sex equality by cultural values. (In complementary analyses we used the GGGI instead. As expected, sex differences in hygiene are not as strongly predicted by the GGGI as by the cultural valuation of sex equality but the overall pattern of results is the same. See S1 and S2 Figs and S2 Table).

## Countries and participants

Hygiene norms were measured in 56 countries as part of ISMN, the International Study of Metanorms [8, 22]. This study aimed at collecting samples of at least 200 students per country and, in most countries, an additional 100 non-students. In the present study of sex differences we include only those participants who also reported information on sex, yielding final sample sizes per country ranging from 45 to 1,009 with a median of 281, summing to a total of 17,632 participants (66.2% women, 33.8% men). As most participants were students (79.9%), the sample was overall quite young (mean age 25.0 years with a standard deviation of 8.9 years). S1 Table presents the exact set of countries and characteristics of each country samples. Measures of attitudes to sex equality (see below) are available for 49 of these countries, comprising three African countries, ten American countries, fourteen Asian countries, twenty-one European countries, and Australia.

The survey (S1 File) was translated into 30 different languages, following the usual practice of independent translation and back-translation. The study was conducted anonymously online using Qualtrics, with a few exceptions. Part of the Estonian non-student sample and the Ghanaian student and non-student samples were collected using pen and paper at the university, with animations shown on a big screen. The survey is available at the Open Science Framework (osf.io/d4q38/).

## Ethics statement

All participants gave informed written consent. All relevant ethical regulations were complied with. Approval of the study protocol was obtained from ethics committees and institutional review boards where required, including: Queen's University (Canada), York University (Canada), Bogotá (Colombia), Institute of Psychology at the Czech Academy of Sciences (Czech Republic), Universidad San Francisco de Quito (Ecuador), United Psychological Research Committee (Hungary), Monk Prayogshala (India), the Trinity College Dublin School of Social Sciences and Philosophy (Ireland), Kwansei Gakuin University (Japan), Aoyama Gakuin University (Japan), United States International University–Africa (Kenya), Sunway University (Malaysia), University of Amsterdam (Netherlands), Komisja ds. Etyki Badań Naukowych Wydziału Psychologii Uniwersytetu SWPS (Poland), Instituto de Ciências Sociais (Portugal), Doha Institute for Graduate Studies (Qatar), Singapore Management University (Singapore), Sungkyunkwan University (South Korea), Universidad de Navarra (Spain), Post Graduate Institute of Medicine (Sri Lanka), Chulalongkorn University (Thailand), American University of Sharjah (United Arab Emirates), University of Kent (United Kingdom), Brunel College of Health and Life Sciences (United Kingdom), University of South Carolina (United States), and New York University (United States).

## Dependent variable: Hygiene norm strictness

The ISMN included 12 items on normative beliefs about spitting and handwashing. Participants were asked where they think it is *not appropriate for people to spit*, with tick boxes for six locations: *in the kitchen sink*, *on the sidewalk*, *on the kitchen floor*, *on the soccer field*, *in the water in a public swimming pool*, and *in the forest*. Participants were also asked in which situations they think people *should wash their hands*, with tick boxes for six situations: *before eating a meal*, *after eating a meal*, *after defecating*, *after urinating*, *when they come home*, and *after shaking hands*. To make aggregated values correspond to percentages, ticks are coded as 100, non-ticks as 0.

### Independent variables: Sex and sex equality

Participants' sex was measured through the question "What is your gender?" with response options Male, Female, and Other/Don't want to say. We analyze only participants who answered Male (coded 0) or Female (coded 1).

To measure the level of *sex equality* in a country we used data from WVS, the World Values Survey [27] and EVS, the European Values Study [28]. Since 1994, these surveys include the EQUALITY index [24], based on three items measuring attitudes to sex equality with respect to jobs, politics, and university education ("When jobs are scarce, men should have more right to a job than women", "On the whole, men make better political leaders than women do", "University is more important for a boy than for a girl"), yielding a measure between 0 and 1. For each country we use WVS /EVS data on sex equality from the most recent wave in which the country participated, yielding data from a total of 166,208 participants in 111 countries, with sample sizes per country ranging from 417 to 3,531. We calculated country means by applying the sampling weights provided with the data. To measure sex equality, we used the country mean of the EQUALITY index. For a few countries there was data on the underlying jobs-related item but not on the full index; we then used the country means for the jobs item in a linear prediction to estimate EQUALITY index scores for these countries. (The rationale for this is that the jobs item is extremely strongly correlated with the full index, $r = 0.97$.) In this way we obtained country measure of sex equality for 49 countries in our study. After multiplication by 100 for convenience, these country measures range from 28 in Saudi Arabia to 93 in Sweden (M = 63, SD = 16).

### Covariates

Following Eriksson et al. [8], we include several variables from the ISMN as covariates. A participant's perceived threat from diseases is measured by whether they ticked "diseases" as a potential threat to their society (coded 1 for tick, 0 for no tick). A participant's valuation of self-control is measured by whether they ticked "feeling of responsibility" as an especially important quality for children to be encouraged to learn at home (coded 1 for tick, 0 for no tick). A participant's response style with respect to tick box questions is measured by the number of ticks they made across nine additional boxes for important child qualities (independence, hard work, imagination, tolerance and respect for other people, thrift, saving money and things, determination/perseverance, religious faith, unselfishness, and obedience). These covariates are also aggregated to country-level variables.

### Alternative moderators

Inspired by Schmitt [16], we also examine whether variation across countries in the sex difference in hygiene norms may be driven by country differences in pathogen prevalence or religiosity rather than by sex equality. We used publicly available country-level data on the historical prevalence of pathogens [29] and religiosity [30].

### Statistical analysis

All analyses below are carried out using SPSS 26.0. The data analyzed here and the code for the analyses behind all tables and figures are all available at the Open Science Framework (osf.io/d4q38/).

Estimates of global sex differences in strictness of norms for each hygiene behavior are obtained from linear mixed-effect models including only intercept and a fixed effect of sex

(dummy for female), in addition to random intercept and random slope of sex at country-level.

In each country we estimate the sex difference in the strictness of each hygiene norm as the coefficient of the dummy variable for female sex in a simple linear regression of the strictness of the norm among all participants in the country.

Correlations are parametric (Pearson's product moment, denoted by *r*) unless explicitly stated otherwise (Spearman's rho).

Our final analyses are mixed-level analyses of the strictness about handwashing and spitting with random intercept and random slope of sex at country level, both without controls and controlling age, student/non-student, response style, perceived threat of disease and valuation of self-control, the latter two variables both at the individual level and aggregated to the country level. The analyses are performed separately in two groups of countries: the 22 countries with below average sex equality and the 27 countries with above average sex equality.

## Results

### Global sex differences in hygiene strictness

As a first step we estimate global sex differences in strictness of hygiene norms by including only intercept and sex as fixed effects. We perform this analysis for each of the twelve norms. Results are reported in Table 1. The intercept is the estimated strictness among men and the effect of sex is the estimated additional strictness among women. Women have stricter norms than men for both behaviors in every context, but the sex difference varies in size across behaviors and contexts. The effect size ranges from negligible (Cohen's $d = 0.03$) to moderately small ($d = 0.30$).

### Sex differences in hygiene strictness per country (Hypothesis 1)

In each country we estimated the national sex difference in the strictness of each hygiene norm as the coefficient of the dummy variable for female sex in a simple linear regression of

**Table 1. Global sex differences in strictness of hygiene norms.**

| Hygiene behavior | Context | Global intercept (men's average strictness) | | Global sex effect (women's additional strictness) | | |
|---|---|---|---|---|---|---|
| | | B | CI | B | CI | d |
| Wash hands | before eating | 86.9 | [84.4, 89.4] | 1.4 | [-1.0, 3.7] | 0.04 |
| | after eating | 50.7 | [45.4, 56.1] | 3.3 | [1.0, 5.6] | 0.07 |
| | after defecating | 91.8 | [89.7, 93.9] | 0.8 | [-0.8, 2.6] | 0.03 |
| | after urinating | 89.5 | [86.8, 92.3] | 1.8 | [-0.0, 4.0] | 0.07 |
| | when they come home | 60.7 | [55.7, 65.8] | 9.1 | [6.8, 11.4] | 0.19 |
| | after shaking hands | 17.1 | [15.1, 19.2] | 2.6 | [0.7, 4.4] | 0.07 |
| Not spit | in the kitchen sink | 53.6 | [49.3, 58.0] | 4.5 | [2.3, 6.7] | 0.09 |
| | on the sidewalk | 64.9 | [60.9, 68.8] | 9.0 | [6.4, 11.6] | 0.20 |
| | on the kitchen floor | 82.1 | [79.6, 84.6] | 3.8 | [2.3, 5.2] | 0.11 |
| | on the soccer field | 37.1 | [33.4, 40.7] | 15.2 | [12.3, 18.2] | 0.30 |
| | in the water in a public swimming pool | 76.8 | [74.0, 79.6] | 5.8 | [3.4, 8.1] | 0.15 |
| | in the forest | 20.9 | [18.5, 23.2] | 10.6 | [7.7, 13.4] | 0.23 |

Estimates of global sex differences in strictness of norms for each hygiene behavior. Note: Results, with 95% confidence intervals, from linear mixed-effect models including only intercept and a fixed effect of sex (dummy for female), in addition to random intercept and random slope of sex at country-level. Cohen's *d* is calculated by dividing the sex effect with the standard deviation in the full sample.

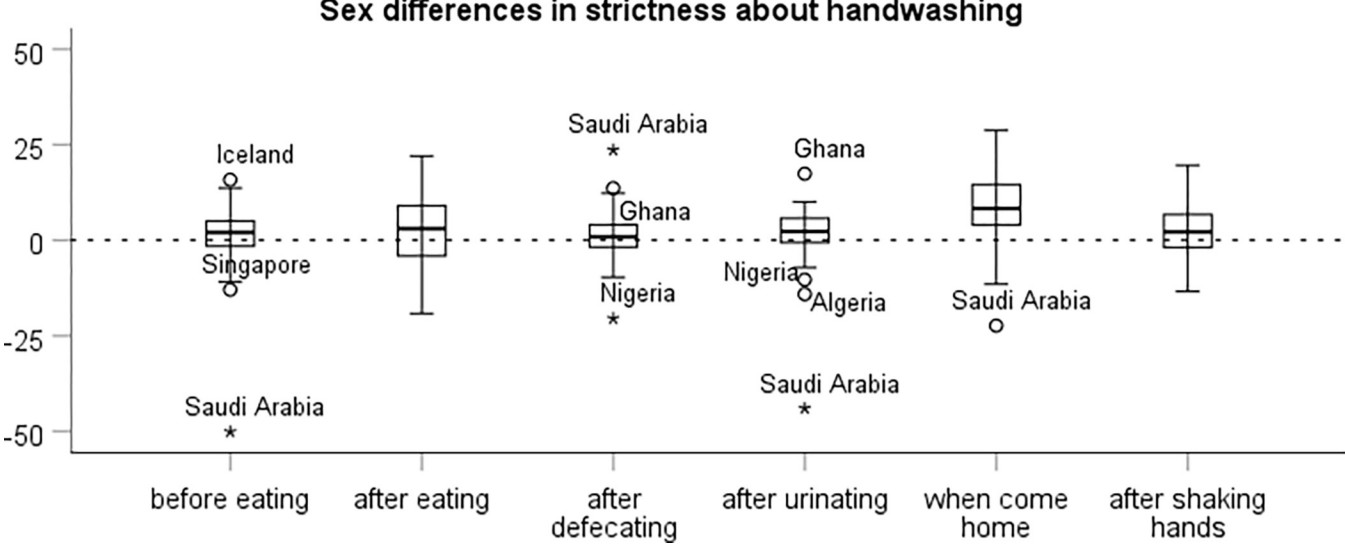

**Fig 1. Sex differences in the strictness of norms about handwashing in different contexts in 56 countries.** The box represents the interquartile range with the dark line indicating the median. The whiskers reach the min and max values in case these are at most 1.5 times the box height outside the interquartile range. Outliers are labeled.

the strictness of the norm among all participants in the country. The results are presented in boxplots in Fig 1 (handwashing) and Fig 2 (spitting). For every norm there are some countries in which the observed national sex difference is negative, that is, where men had somewhat stricter norms than women. However, for the clear majority of countries we observe a sex difference in strictness in the expected direction. Some of the aberrant results are likely due to noise. A way to reduce the noise is to aggregate national sex differences in hygiene norms across contexts. However, aggregation across contexts is a meaningful way to reduce noise

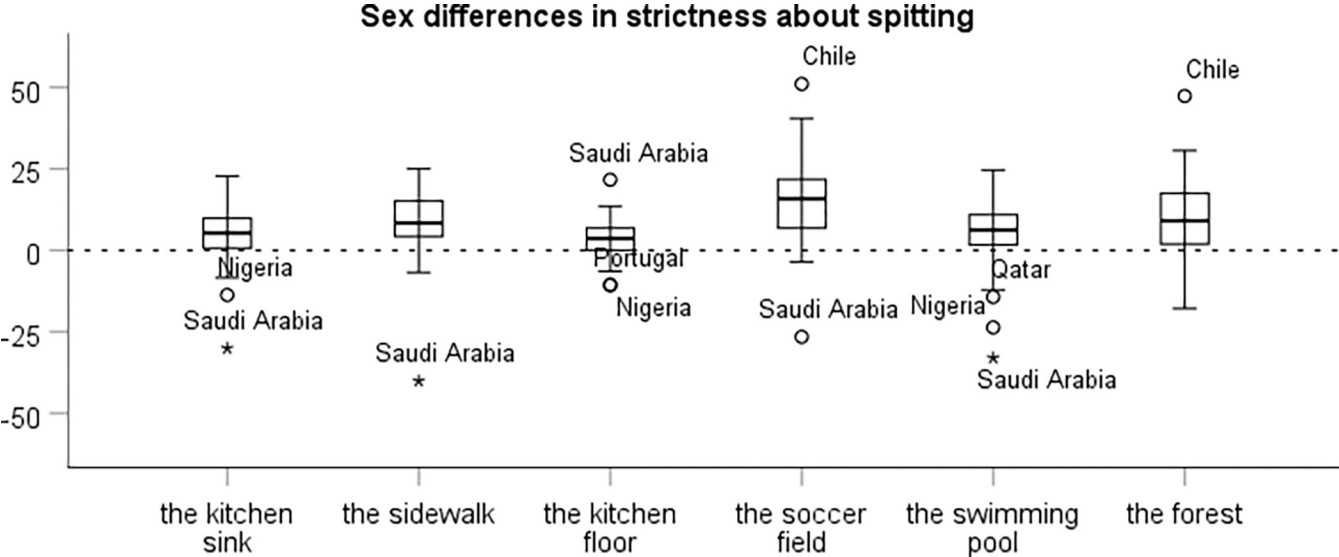

**Fig 2. Sex differences in the strictness of norms about spitting in different contexts in 56 countries.** The box represents the interquartile range with the dark line indicating the median. The whiskers reach the min and max values in case these are at most 1.5 times the box height outside the interquartile range. Outliers are labeled.

only if there is a consistent underlying pattern in national sex differences in hygiene norms. We therefore first need to check that national sex differences in hygiene norms in different contexts indeed exhibit a sufficient level of consistency.

**Internal consistency of the national sex difference in handwashing norms.** The internal consistency of the national sex difference in the six handwashing norms is adequate, $\alpha = .78$, indicating that it is meaningful to aggregate national sex differences in handwashing norms across contexts. (Note that this is a separate question from whether it is meaningful to aggregate individual ratings across contexts, which is addressed in the mixed-level analysis below.) However, closer inspection reveals that the national sex difference in the norm about handwashing after defecating is uncorrelated with the national sex difference in the other handwashing norms (perhaps due to a ceiling effect). If this norm is excluded, the internal consistency increases to $\alpha = .84$. For this reason, we use the average national sex difference in the remaining five handwashing norms as a country index of the sex difference in strictness about handwashing (M = 3.3, SD = 6.7). However, all results reported below are similar if we instead base the index on all six handwashing norms.

**Internal consistency of the national sex difference in spitting norms.** The case of spitting norms follows a similar pattern. The internal consistency of the national sex difference in the six spitting norms is adequate, $\alpha = .73$, but closer inspection reveals that the national sex difference in the norm about spitting on the kitchen floor is uncorrelated with the national sex difference in the other spitting norms. If this norm is excluded, the internal consistency increases to $\alpha = .80$. We use the average national sex difference in the remaining five spitting norms as a country index of the sex difference in strictness about spitting (M = 8.7, SD = 7.8), but all results reported below are similar if we instead base the index on all six spitting norms.

**Aggregated sex differences in hygiene strictness per country.** The two sex difference indices per country are shown in Fig 3. In line with Hypothesis 1, the great majority of countries, 45 out of 56, are in the positive quadrant. In the negative quadrant we find two countries: Saudi Arabia and Nigeria. Our study thus indicates that in these two countries, norms about spitting and handwashing are stricter among men than among women. For handwashing, a considerable negative sex difference in strictness is also observed in Algeria, Singapore, and Botswana.

There are additional noteworthy features of the data that Fig 3 illustrates. National sex differences in hygiene strictness are clearly correlated across handwashing and spitting, $r = .69$ (or $r = .44$ if the outlier Saudi Arabia is excluded). Thus, there is a domain general pattern of national variation in sex differences in hygiene strictness. It is further worth noting that the sex difference is typically larger for spitting than for handwashing; this held in 47 out of 56 countries.

## Relation between the sex difference in hygiene strictness and sex equality (Hypothesis 2)

We now turn to the question of whether the sex difference in hygiene strictness in a country is related to its level of sex equality. Because Saudi Arabia is such an outlier, we use Spearman's correlation instead of Pearson's. Results are qualitatively similar.

**Handwashing.** In sharp contrast to expectations (Hypothesis 2), the sex difference in strictness is *positively* correlated with sex equality, Spearman's rho = 0.45, $p = 0.001$, $n = 49$. However, a scatter plot (Fig 4) shows that this is not a linear increase. Instead, there is a steep increase from the most inegalitarian country (Saudi Arabia, with sex equality value 28) to countries with average levels of equality, at which point the sex difference in strictness about handwashing reaches a plateau. If we split the countries in two groups, below and above

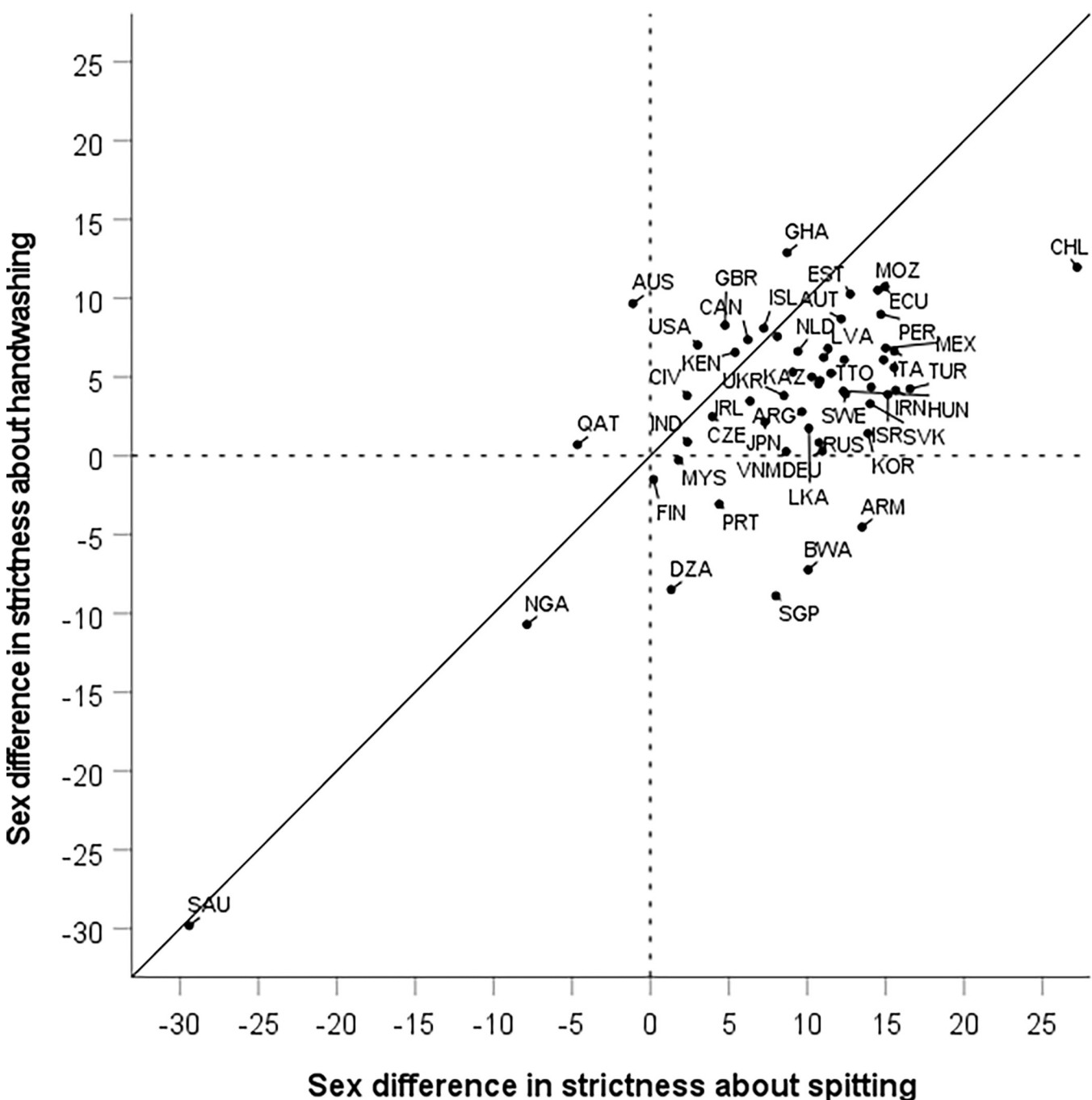

**Fig 3. The sex difference in strictness about handwashing (average of five items) plotted against the sex difference in strictness about spitting (average of five items) in 56 countries shows that both differences are positive in the great majority of countries.** Saudi Arabia (SAU) and Nigeria (NGA) are clear exceptions with both differences negative; moreover, Algeria (DZA), Singapore (SGP), and Botswana (BWA) are clear exceptions with respect to handwashing but not with respect to spitting. Labels are ISO country codes.

average sex equality, drawing the line between Ecuador (ECU) and Bosnia and Herzegovina (BIH), the correlation between sex equality and strictness about handwashing in the first group is strongly positive, Spearman's rho = 0.54, $p$ = 0.010, $n$ = 22, while in the second group there is no correlation, Spearman's rho = 0.09, $p$ = 0.65, $n$ = 27.

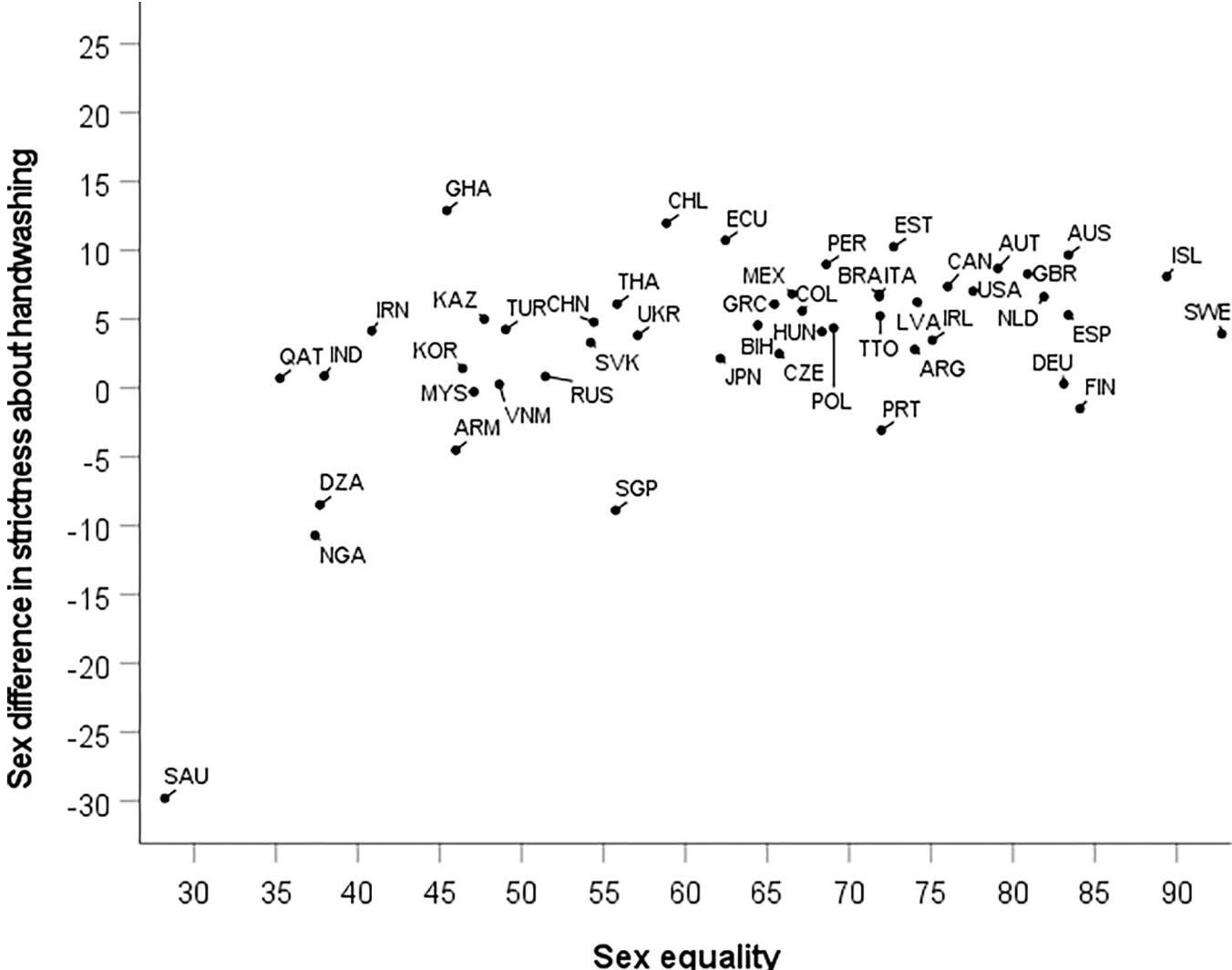

**Fig 4. National sex differences in strictness about handwashing plotted against a measure of (cultural valuation of) sex equality from the WVS/EVS in 49 countries.** Labels are ISO country codes.

**Spitting.**   The sex difference in strictness about spitting is uncorrelated with sex equality, Spearman's rho = 0.02, *p* = 0.88, *n* = 49. However, a scatter plot (Fig 5) reveals that the two variables are in fact strongly non-linearly related. Among countries with below-average levels of sex equality, the correlation between sex equality and strictness about spitting is strongly positive, Spearman's rho = 0.52, *p* = 0.013, *n* = 22. Among countries with above-average levels of sex equality, the correlation is equally strongly negative, Spearman's rho = -0.51, *p* = 0.006, *n* = 27.

## Robustness to adjustment for covariates in a mixed-level model

So far, we have only considered the raw sex difference without any adjustment for other factors that are known to impact hygiene norms, including individual-level factors. To examine the robustness of our findings to such factors, we examine if our findings replicate in analysis of individual-level data when controls are included. To mirror the country-level analysis, we

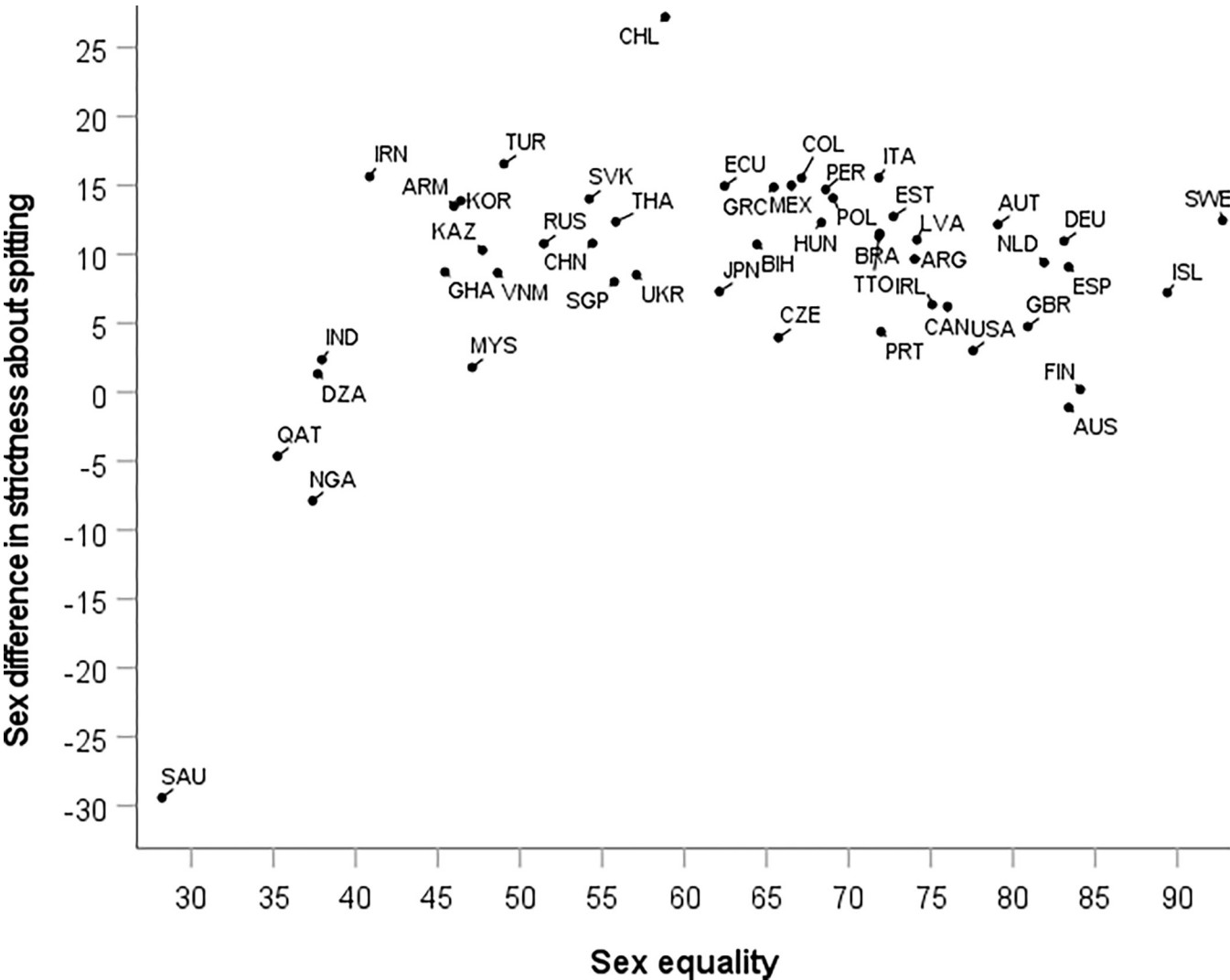

**Fig 5. National sex differences in strictness about spitting plotted against a measure of (cultural valuation of) sex equality from the WVS/EVS in 49 countries.** Labels are ISO country codes.

average the individual's dichotomous responses to handwashing in five contexts, α = .51. We similarly average the individual's dichotomous responses to spitting in five contexts, α = .58. (These values of Cronbach's alpha are on the low side, which tends to be the case when data are dichotomous). We perform mixed-level analyses of the individual-level data on strictness about handwashing and spitting with random intercept and random slope of sex at country level. We compare the results without controls to results when including controls for age, student/non-student, response style, perceived threat of disease and valuation of self-control, the latter two variables both at the individual level and aggregated to the country level. The analyses are performed separately in two groups of countries: the 22 countries with below average sex equality and the 27 countries with above average sex equality.

Table 2 reports the results for the individual-level dummy for female sex (i.e., the sex effect) and its interaction with country-level sex equality, centered on the mean in each group of countries so that the sex effect refers to the average in the group of countries. First note that these multi-level analyses replicate the findings in the previous country-level analyses. Thus,

**Table 2. Strictness about handwashing and spitting.**

| Variable | Strictness about handwashing | | | | Strictness about spitting | | | |
|---|---|---|---|---|---|---|---|---|
| | Countries w. *below* average sex equality | | Countries w. *above* average sex equality | | Countries w. *below* average sex equality | | Countries w. *above* average sex equality | |
| | w/o cont. | w. cont | w/o cont. | w. cont | w/o cont. | w. cont | w/o cont. | w. cont |
| Female | 1.6 | 0.5 | 5.5 [a] | 4.9 [a] | 8.9 [a] | 8.0 [a] | 9.8 [a] | 9.3 [a] |
| Female × Sex equality | 0.6 [b] | 0.7 [a] | -0.0 | -0.0 | 0.8 [a] | 0.8 [a] | -0.3 [c] | -0.3 [c] |
| Random effects | | | | | | | | |
| Variance of individual residuals | 538.3 | 506.5 | 454.3 | 445.1 | 666.8 | 652.4 | 683.7 | 677.9 |
| Variance of random intercepts | 41.8 | 27.9 | 22.2 | 14.5 | 75.2 | 51.3 | 42.9 | 37.2 |
| Variance of random slopes of sex | 44.6 | 51.7 | 2.1 | 1.2 | 57.1 | 58.6 | 8.8 | 8.2 |
| N (individuals) | 6728 | 6728 | 9350 | 9350 | 6728 | 6728 | 9350 | 9350 |
| N (countries) | 22 | 22 | 27 | 27 | 22 | 22 | 27 | 27 |
| BIC | 62798 | 61093 | 83630 | 87583 | 62976 | 62798 | 87690 | 87583 |

Results from analysis strictness about handwashing and spitting using linear mixed-effect models. Note: Entries are unstandardized coefficients. Models "without controls" included intercept and sex equality. Models "with controls" additionally included age, dummy for student, response style, perceived threat of disease and valuation of self-control (the latter two variables both at the individual level and aggregated to the country level).

[a]: $p < .001$

[b]: $p < .01$

[c]: $p < .05$.

the sex effect on strictness about handwashing increases with sex equality in the below-average group of countries, but it does not change with sex equality in the above-average group; the sex effect on strictness about spitting increases with sex equality in the below-average group of countries and decreases with sex equality in the above-average group. Second, note that these results are unchanged when controls are included. Even the average sex effects are essentially unchanged when the controls are added. Thus, sex differences in hygiene strictness are not accounted for by any sex differences in participants' age, response style, perceived threat of disease, or valuation of self-control.

**Alternative moderators.** Finally, we examine whether country variation in sex differences in hygiene norms could be explained by pathogen prevalence or religiosity instead of sex equality [16]. To account for non-linearity in a comparable way across moderators, we use linear regression including a quadratic term in addition to a linear term. In such quadratic models, sex equality accounts for large proportions of the variation in national sex differences in hygiene strictness: 46% in the case of handwashing and 48% in the case of spitting. The alternative moderators fare much less well. Quadratic models that use religiosity instead only accounts for 5% and 14% of the variation, respectively. Quadratic models that use pathogen prevalence instead only accounts for 5% and 2% of the variation, respectively.

## Discussion

Using data in 56 countries, the current study provided a comprehensive analysis of the sex difference in hygiene strictness. We studied a set of norms about when you should wash your hands and where you should not spit. Globally, we found norms about handwashing to be slightly stricter among women than among men. The direction of this sex difference is consistent with findings in many single-country behavioral studies of handwashing [11–15]. We found even more substantial sex differences in the strictness of spitting norms. This is an important novel finding as no prior studies have examined sex differences in spitting. Perhaps it is related to men producing more saliva than women do [31], as this might create a stronger

preference in favor of spitting. It is possible that spitting elicits a stronger disgust response in women by association because spitting is an innate behavioral response to remove noxious material from the mouth [32]. But there are also differences in the types of norms the two behaviors are involved in. For one thing, handwashing norms are prescriptive while norms about spitting are proscriptive. For another, handwashing is primarily a private good while strictness about spitting is primarily a public good. Moreover, the handwashing norms we studied were concerned with *when* you should wash your hands whereas the spitting norms were concerned with *where* you should not spit. This could play a role as culture may restrict women's access to certain locations (e.g., soccer pitches). Future work may examine the specific roles of these factors.

Our first hypothesis was that the sex difference in hygiene strictness would be observed everywhere. In our dataset, we observed the sex difference in most countries but not all. Two countries, Saudi Arabia and Nigeria, were clear exceptions in that men reported stricter hygiene norms than women. To validate this finding, we searched for prior studies of hygiene in any of these countries that report results separately for men and women. We found two such studies for Saudi Arabia, both of which indeed reported stricter hygiene among men than among women [33, 34]. Thus, our findings are consistent with prior literature. We conclude that sex difference in hygiene strictness is nearly universal but that the presence of clear exceptions demonstrates that there is some cultural moderator that needs to be understood.

Our second hypothesis examined a proposed cultural moderator: the level of sex equality in society. We operationalized societal sex equality by the average attitude to sex equality with respect to participation in the job market, in politics, and in higher education. Such attitudes can be taken as a proxy for the strength, or weakness, of sex roles. Some authors have attributed stricter hygiene norms among women to sex roles [2, 21]. However, a more complex picture emerged in our data. The sex difference in hygiene strictness was often larger in countries with above-average levels of sex equality than in countries with below-average levels of sex equality. This finding is in keeping with many behavioral and somatic sex difference results and Schmitt has argued that biological sex differences can be moderated and facilitated by specific cultural contexts [16]. To our knowledge it has not be shown for the contents of normative beliefs. Within these groups of countries, the sex difference in hygiene strictness varied with the level of sex equality in different ways. Among countries with above-average sex equality, the sex difference in handwashing strictness showed no relation with sex equality whereas the sex difference in spitting strictness showed a negative relation with sex equality. Thus, the hypothesis was partially supported in this group of countries. Results looked very different in the group of countries with below-average levels of sex equality. In this group, the expected sex difference was most pronounced at the high end of sex equality, that is, at global average levels of sex equality. In countries with greater inequality, the sex difference in hygiene strictness disappeared and even became reversed at extreme levels of inequality (Saudi Arabia and Nigeria). This reversal suggests a flexible connection, if any, between hygiene norms and established sex differences in disgust sensitivity. But it is possible that the underlying asymmetry in inclusive fitness costs is something that can drive either male or female custodianship of hygiene, and hence difference in hygiene strictness, dependent upon key social ecological factors. Our analyses strongly supported that sex equality is a key factor. Specifically, while prior research has found that pathogen prevalence and religiosity may be more important factors behind sex differences in other domains [16], we found the level of sex equality do be a much stronger predictor of sex differences in hygiene.

In sum, we have found that there is substantial cultural variability in the extent to which women have stricter hygiene norms than men do, and it is quite strongly related to sex equality—but in a non-linear way. A possible interpretation of this unexpected finding is that the

full spectrum of sex inequality encompasses several distinct phenomena. If we hold on to the notion that the sex difference is due to sex roles giving women a greater responsibility for maintaining hygiene in society, how could this responsibility vary across different levels of gender inequality? In moderately unequal societies, both women and men are fully responsible, but they tend to have different responsibilities. Women and men are seen as working together in a family unit where she is responsible for raising the children and keeping the home clean while he is responsible for bringing home most of the income. It is in these societies we would expect women to be more responsible for hygiene. As societies become more egalitarian, these sex roles weaken, and we would expect a decline in the sex difference in hygiene. In the most unequal societies, however, it is arguably men that have the responsibility in that they make decisions about the whole family's behavior and are held responsible for the behavior of their wives or daughters. As a case in point, all women in the extremely unequal society of Saudi Arabia have a legal male guardian who is responsible for them [35]. Among other things, this ultra-low level of women's responsibility means a lower level of responsibility for hygiene. Sex segregation in Saudi Arabia also implies that women are allowed less mobility and thereby potentially less exposure to situations and things that may motivate hygiene norms, in this way moderating any underlying biological sex differences [16]. This could be examined in future research.

**Limitations.**    Limitations of the data were discussed by Eriksson et al. [8]. Most importantly, as the data are limited to hand washing and spitting norms, we cannot say whether the sex difference in hygiene strictness generalizes to other hygiene-related behaviors, such as washing the whole body, washing clothes, coughing, sneezing, and urinating. Other limitations include that African countries and small countries were undersampled, that socioeconomic stratification within countries is not measured, and that samples per country are sometimes quite small and not necessarily representative. However, these may not be major concerns, as prior analyses of data from these samples successfully replicate country-level variation in cultural values found in representative samples [22, 36]. Another limitation is that we do not have data on participants' knowledge of objective benefits of hygiene.

## Conclusion

We found a near universal sex difference in hygiene strictness in a large cross-cultural sample. What underlies this sex difference? Those who seek biological explanations would have to find a way to explain the large and systematic variation between countries. On the other hand, those who seek sociological explanations would have to explain why the link between gender and stricter hygiene norms have developed in so many places and why the exceptions are found in the most unequal societies. We have suggested some explanations but at this point they remain speculative. The global pattern of sex differences in hygiene norms that we have demonstrated empirically, points to a need for further theory development and more detailed data.

## Supporting information

**S1 Fig. National sex differences in strictness about handwashing plotted against the Global Gender Gap Index (GGGI).** GGGI data for 56 countries taken from the World Economic Forum's Global Gender Gap Report for 2020. Labels are ISO country codes.
(TIF)

**S2 Fig. National sex differences in strictness about spitting plotted against the Global Gender Gap Index (GGGI).** GGGI data for 56 countries taken from the World Economic Forum's

Global Gender Gap Report for 2020. Labels are ISO country codes.
(TIF)

**S1 Table. Sample characteristics per country.**
(DOCX)

**S2 Table. Mixed-level analyses of the strictness about handwashing and spitting using the Global Gender Gap Index (GGGI) as a measure of sex equality.**
(DOCX)

**S1 File. Full text of the ISMN survey.**
(DOCX)

## Author Contributions

**Conceptualization:** Kimmo Eriksson, Thomas E. Dickins, Pontus Strimling.

**Data curation:** Kimmo Eriksson.

**Formal analysis:** Kimmo Eriksson.

**Funding acquisition:** Kimmo Eriksson.

**Project administration:** Kimmo Eriksson.

**Writing – original draft:** Kimmo Eriksson.

**Writing – review & editing:** Kimmo Eriksson, Thomas E. Dickins, Pontus Strimling.

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
