## [Decision Letter · Decision Letter 0]

11 Jan 2022

PGPH-D-21-00958

Global sex differences in hygiene norms and their relation to sex equality

Dear Dr. Eriksson,

Thank you for submitting your manuscript to PLOS Global Public Health. After careful consideration, we feel that it has merit but does not fully meet PLOS Global Public Health’s publication criteria as it currently stands. Therefore, we invite you to submit a revised version of the manuscript that addresses the points raised during the review process.

We look forward to receiving your revised manuscript.

Kind regards,

Mahbub-Ul Alam, MPH

Academic Editor

Journal Requirements:

1. Please include additional information regarding the survey or questionnaire used in the study and ensure that you have provided sufficient details that others could replicate the analyses. For instance, if you developed a questionnaire as part of this study and it is not under a copyright more restrictive than CC-BY, please include a copy, in both the original language and English, as Supporting Information.

2. Please amend your detailed Financial Disclosure statement. This is published with the article, therefore should be completed in full sentences and contain the exact wording you wish to be published.

State what role the funders took in the study. If the funders had no role in your study, please state: “The funders had no role in study design, data collection and analysis, decision to publish, or preparation of the manuscript.”

3. Please note that Figure 1 is currently split into multiple parts in separate files. Please either combine the parts into one figure with lettered panels, or renumber your figures as whole parts (i.e. Fig 1A becomes Fig 1, Fig 1B becomes Fig 2 etc.), ensuring that all relevant legends and in-text citations have been updated accordingly.

Additional Editor Comments (if provided):

Reviewers' comments:

Reviewer's Responses to Questions

**Comments to the Author**

1. Does this manuscript meet PLOS Global Public Health’s publication criteria? Is the manuscript technically sound, and do the data support the conclusions? The manuscript must describe methodologically and ethically rigorous research with conclusions that are appropriately drawn based on the data presented.

Reviewer #1: Yes

Reviewer #2: Yes

2. Has the statistical analysis been performed appropriately and rigorously?

Reviewer #1: Yes

Reviewer #2: Yes

3. Have the authors made all data underlying the findings in their manuscript fully available (please refer to the Data Availability Statement at the start of the manuscript PDF file)?

Reviewer #1: Yes

Reviewer #2: Yes

4. Is the manuscript presented in an intelligible fashion and written in standard English?

Reviewer #1: Yes

Reviewer #2: Yes

5. Review Comments to the Author

Reviewer #1: Global sex difference in hygiene norms and their relation to sex equality is a noble study, given the role of hygiene in dairrheal and other infectious diseases spread worldwide.

This study demonstrate sex differences in terms of stricter hygiene norms; however, it would be of interest to also look at some of the explanatory variables for this observed universal sex difference in hygiene strictness in a large cross-cultural sample.

What is not clear, the sample size distribution per country was based on? an whether they would have an influence on the statistical outcome.

.

Reviewer #2: This paper describes data collected by the International Study of Metanorms project. In this study, the authors and their collaborators assessed the endorsements of a variety of norms across 56 nations. Relevant to the current paper, some of these norms concerned acts that can transmit pathogens – specifically, handwashing and spitting. The current manuscript reports that, across cultures, women were more likely (1) to find it inappropriate to spit and (2) to believe that people should wash their hands. A nation-level variable assessing endorsements of sex equality moderated this sex difference in a non-linear fashion.

I think that this is a thought-provoking paper that would contribute nicely to multiple literatures. However, I think that some of the analyses should be better justified and potentially redone. I’ll provide a list of suggestions below.

ISSUE 1: OPERATIONALIZTION OF SEX EQUALITY.

The authors refer to sex equity in the manuscript title, abstract, and text, and they also refer to egalitarianism. But the variable they use to operationalize this construct assesses attitudes toward sex equality among citizens of those countries, as estimated from the World Values Survey (e.g., “University is more important for a boy than a girl”). This decision is justified as follows: “When examining causal effects of sex equality, it is preferable to use a belief measure instead of an outcome measure such as the Global Gender Gap Index; as other researchers has (sic) pointed out, the same outcome may be achieved in different ways.”

I’m not persuaded by this justification, and I doubt that the majority of readers will be. After all, the argument presented in the introduction doesn’t concern attitudes; it concerns structural differences in the roles that the sexes typically fill.

Ultimately, there’s little harm in also reporting how a more objective measure of sex equality moderates the sex differences reported in the manuscript. Differences in outcomes across the moderators could be useful in generating further hypotheses explaining these sex differences. I’d also prefer for the variables to be labeled in a more precise manner. For example, the WVS measure could be labeled “Sexist Attitudes,” whereas the Global Gender Index could be labeled “Sex Equality.”

ISSUE 2: DATA AGGREGATION

I found the description of data aggregation (lines 237-252) confusing. The manuscript states “The internal consistency of the sex difference in the six handwashing norms is adequate (alpha = .78).” These analyses are either reported imprecisely, or they reflect an assessment of reliability that I’m unfamiliar with (and that departs markedly from standard practices). Coefficient alpha is based on the number of items and the average covariances between items, and it is an estimate of the proportion of observed variance attributable to variance in the construct of interest. A researcher might be interested in assessing the reliability of six handwashing norm items; this reliability would be computed as: (1) the number of items times the average interitem correlation divided by (2) one plus N minus 1 times the average interitem correlation. Estimating reliability in this manner and assigning each participant a single value of handwashing norm endorsement is a reasonable strategy for both reducing the observations that need to be analyzed and obtaining an estimate that less composed of idiosyncratic item-level variance.

But the phrase “internal consistency of the sex difference” is puzzling. I’m not sure if the phrasing is imprecise, or if the analyses have been conducted at the national level, with each nation having an average sex difference across participants per item. This latter approach would be a problem. Variables can have similar sex differences at an aggregate level, but be uncorrelated at a lower level (this is analogous to the ecological fallacy).

In sum, I’d urge either increased precision in the language used to describe these reliability estimates, or I’d urge reliability to be assessed at the item-level rather than at the country-level.

(One other related note on this point: the authors have usefully included analysis script on the OSF, but the script for these reliability analyses was not included. I’d urge them to include this script, just to avoid confusion from other readers)

ISSUE 3: TREATMENT OF SAUDI ARABIA

The findings from Saudi Arabia are fascinating. But I’d urge the authors to do two things. First, double check the data. In a past project, I organized data collection across 30 nations. The mean for one of the critical variables in this study was higher for women than for men in 28 of the 30 nations (more about this variable below). The sex difference was almost identical in magnitude, but in the opposite direction, for the other two nations. When I asked the researchers from those two nations to verify that they had coded for participant sex in the correct manner, they realized that they had made a mistake. Hence, the reversed sex difference was caused by a coding error.

Second, if the pattern remains after verifying that the data are correct, I’d encourage a bit more reflection on why that pattern might exist. Consider, for example, the role of hygiene (especially handwashing) in Islamic prayer rituals (e.g., Wadu). Ritualized hygiene practices might attenuate the sex difference; ritualized hygiene practices combined with some sex segregation in these practices might reverse them. Naturally, these thoughts are speculative. Nevertheless, speculation can lay the ground for future research.

ISSUE 4: LASER FOCUS ON SOCIAL ROLES

As noted earlier, I think that this work is very interesting, and I suspect that others will feel the same. I do think it could be a bit broader, though. Consider the project I alluded to above (http://www.pnas.org/cgi/doi/10.1073/pnas.1607398113). We assessed pathogen disgust sensitivity across 30 nations. That is, we asked participants to report how disgusting they find stepping on dog poop, seeing mold on food, sitting close to someone with red sores on their arms, etc. The mean for women was higher than the mean for men in every one of these 30 nations. Now, perhaps this sex difference also emerges from the systemic differences in social roles across the sexes. Though I doubt it. A large body of work has mostly not supported the idea that such structural differences explain sex differences in personality, mate preferences, occupational goals, etc. See DOI:10.1007/978-3-319-09384-0_11 Table 11.1 for a succinct summary of these efforts.

Ultimately, the structural hypotheses that the authors test are definitely worth testing! But they’re not the only game in town for explaining sex differences. And there’s a lot of work assessing and explaining sex differences in emotions and behaviors that neutralize pathogens. Some greater engagement with this work might strengthen the manuscript. I’ll provide a few further references to consider.

https://doi.org/10.1007/978-1-4939-0314-6_15

https://doi.org/10.1177/1754073917709940

https://doi.org/10.1016/j.paid.2011.04.003

I hope that these comments are useful, and I wish the authors the best.

6. PLOS authors have the option to publish the peer review history of their article (what does this mean?). If published, this will include your full peer review and any attached files.

**Do you want your identity to be public for this peer review?** For information about this choice, including consent withdrawal, please see our Privacy Policy.

Reviewer #1: No

Reviewer #2: **Yes: **Joshua Tybur

---

## [Decision Letter · Decision Letter 1]

22 Mar 2022

PGPH-D-21-00958R1

Global sex differences in hygiene norms and their relation to sex equality

Dear Dr. Eriksson,

Thank you for submitting your manuscript to PLOS Global Public Health. After careful consideration, we feel that it has merit but does not fully meet PLOS Global Public Health’s publication criteria as it currently stands. Therefore, we invite you to submit a revised version of the manuscript that addresses the points raised during the review process.

We look forward to receiving your revised manuscript.

Kind regards,

Mahbub-Ul Alam, MPH

Academic Editor

Journal Requirements:

Additional Editor Comments (if provided):

Thanks authors for the revision. Please address the below comments from Reviewer 2:

I think that the authors have done a very nice job of revising their manuscript, and I endorse its publication. However, I think that the manuscript still contains a small but important problem, and it still could do a better job of integrating with the cross-cultural disgust literature. The second issue is one of preference on their part; the first issue, I think, is something that is simply wrong and should be amended for their sake.

First issue: the continued reporting of alpha of sex differences. Alpha is used to assess the reliability of a group of items. It's based on assumptions that each item in a group of items loads identically on a latent construct, and it estimates the degree to which the sum of those items reflects latent variable variance (relative to non-systematic or error variance). The construct of interest here is hygiene behaviors, and the authors are examining sex differences in that construct (and, further, seeing how those sex differences vary across a number of countries). Reporting alpha in the way that the manuscript continues to do is a problem for two reasons. First, by taking national averages, it removes information regarding individual-level responses. Second, and relatedly, by examining sex differences, it removes any information about how the items correlate with each other. Variables can have similar sex differences without relating to each other within the sexes. Again, please refer back to myriad examples of the ecological fallacy.

This is a straightforward issue to correct. Just report individual-level alphas (and, ideally, do so separately for each country in the supplement). The lower alpha at the individual level isn't necessarily a problem - it simply informs the degree to which the composites are noisy. I really do urge the authors to follow this suggestion - I think they're making a serious error by using alpha in the way that they use it, and they risk becoming yet another example of myriad inappropriate uses of coefficient alpha.

Second issue: I appreciated the inclusion of pathogen prevalence and some comments on disgust. I do think that the authors could say a bit more about these topics and link their paper with findings from this paper (https://doi.org/10.1073/pnas.1607398113), which (1) demonstrates cross-culturally stable sex differences in disgust (something that none of the papers currently cited on line 71 do, though the manuscript currently implies that they do), and (2) similarly finds a weak-to-null relation with pathogen prevalence (though the the paper I'm referring to examines pathogen disgust across the sexes rather than the sex difference). Again, this is the authors' choice. But I'd think that linking the current cross-cultural paper on hygiene norms to this cross-cultural paper on disgust (and its data on sex differences) would strengthen their contribution.

Reviewers' comments:

Reviewer's Responses to Questions

**Comments to the Author**

1. If the authors have adequately addressed your comments raised in a previous round of review and you feel that this manuscript is now acceptable for publication, you may indicate that here to bypass the “Comments to the Author” section, enter your conflict of interest statement in the “Confidential to Editor” section, and submit your "Accept" recommendation.

Reviewer #1: All comments have been addressed

Reviewer #2: All comments have been addressed

2. Does this manuscript meet PLOS Global Public Health’s publication criteria? Is the manuscript technically sound, and do the data support the conclusions? The manuscript must describe methodologically and ethically rigorous research with conclusions that are appropriately drawn based on the data presented.

Reviewer #1: Yes

Reviewer #2: Yes

3. Has the statistical analysis been performed appropriately and rigorously?

Reviewer #1: Yes

Reviewer #2: No

4. Have the authors made all data underlying the findings in their manuscript fully available (please refer to the Data Availability Statement at the start of the manuscript PDF file)?

Reviewer #1: Yes

Reviewer #2: Yes

5. Is the manuscript presented in an intelligible fashion and written in standard English?

Reviewer #1: Yes

Reviewer #2: Yes

6. Review Comments to the Author

Reviewer #1: The authors have adequately addressed the issues raised by reviewers and I believe it warranty consideration for publication.

Reviewer #2: (No Response)

7. PLOS authors have the option to publish the peer review history of their article (what does this mean?). If published, this will include your full peer review and any attached files.

**Do you want your identity to be public for this peer review?** For information about this choice, including consent withdrawal, please see our Privacy Policy.

Reviewer #1: No

Reviewer #2: No

---

## [Editor Report · Decision Letter 2]

12 May 2022

Global sex differences in hygiene norms and their relation to sex equality

PGPH-D-21-00958R2

Dear Prof. Eriksson,

We are pleased to inform you that your manuscript 'Global sex differences in hygiene norms and their relation to sex equality' has been provisionally accepted for publication in PLOS Global Public Health.

Best regards,

Julia Robinson

Staff Editor
